# Fault Prediction Model of High-Power Switching Device in Urban Railway Traction Converter with Bi-Directional Fatigue Data and Weighted LSM

**Lei Wang** [1,*] **, Shenyi Liu** [1]**, Ruichang Qiu** [2] **and Chunmei Xu** [1]

[1]   School of Electrical Engineering, Beijing Jiaotong University, Beijing 100044, China;
      18126108@bjtu.edu.cn (S.L.); chmxu@bjtu.edu.cn (C.X.)
[2]   Beijing Electrical Engineering Technology Research Center, Beijing Jiaotong University, Beijing 100044,
      China; rchqiu@bjtu.edu.cn
*    Correspondence: leiwang@bjtu.edu.cn

**Abstract:** The switching device is relatively weakest in the traction converter, and this paper aims at the fault prediction of it. Firstly, the mathematical distribution is analyzed based on the results that were obtained in electro thermal simulation and a single-directional accelerated fatigue test. Then, the accelerated fatigue test with bi-directional fatigue current is proposed, the data from which reflects the accelerating effect from FWD on the device aging process. The analytical model of fatigue process is fitted with the data that were obtained in the test. In order to shorten the test time consumption, we propose a weighted least squares method (LSM) to fit the failure data. Finally, the prediction model is presented with the consideration of fatigue signature and Arrhenius temperature factor.

**Keywords:** fault prediction; fatigue model; accelerated fatigue test; high-power switching device

---

## 1. Introduction

The urban rail transit system is a reliable and environmentally friendly way of transport; it generates low pollution, has high passenger capacity and is not prone to delays. The traction converter (TC) in the EMU provides motive and electric braking power through traction motors. TC's switching IGBTs are high-power devices that are prone to failure compared to other components [1]. Switching device fault accounts for more than 10% of the total TC failures. In most cases, fatigue causes the failure of the switching device [2]. If the fatigue state of the switching device can be predicted and manually maintained or replaced in the device [3], the train failure caused by the failure of the switching device can be reduced.

The switching device in the TC is usually a package module composed of an IGBT and an FWD. The fatigue of the switching device is mainly manifested by package failure, including two main forms: Wire liftoffs and solder layer cracking [4,5]. Wire liftoffs and solder layer cracking are caused by shear thermal stresses that are produced by temperature drift inside the device. In the intermittent operation of the device when it is turned on, heat is generated by current; when it is turned off, i.e., the current is cut off, the heat is dissipated. In such a way, the device is heated and cooled periodically. The internal temperature of the device (not only the IGBT junction temperature $T_j$, however also the temperature of each single layer) would increase and decrease accordingly. One switching device consists of multiple layers (as shown in Figure 1) with different coefficients of thermal expansion (CTE) of each layer.

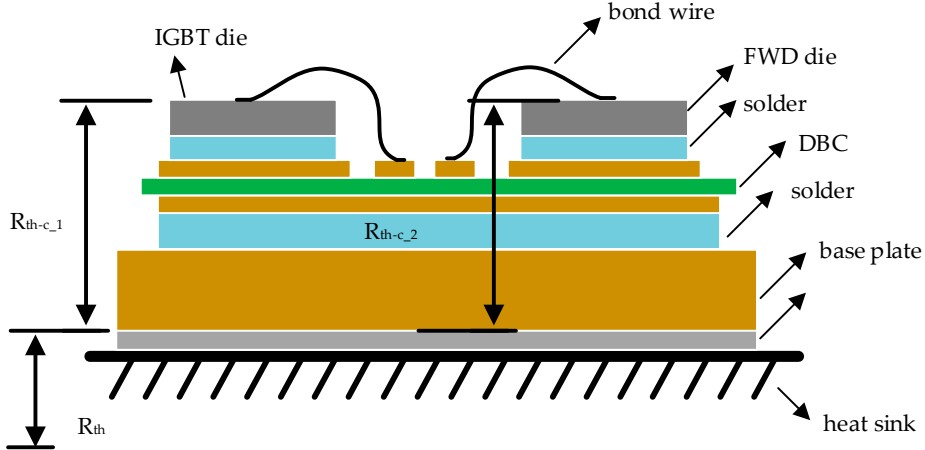

**Figure 1.** Internal structure of the switchgear.

The relationship between thermal stress and the change of CTE is shown in Equation (1).

$$S = f(\Delta CTE, \Delta T) \tag{1}$$

$S$ is the thermal stress, $\Delta CTE$ is the difference of $CTE$ and $\Delta T$ is junction temperature variation. Equation (1) shows that the thermal stress is caused by the difference of $CTE$ and $\Delta T$.

Such temperature change is caused by internal thermal resistance (such as $R_{th-c\_1}$ and $R_{th-c\_2}$ in Figure 1) and heat flow, as is shown in Equation (2)

$$\Delta T = P \bullet Rth \tag{2}$$

In Equation (2), $P$ is the heating power. $R_{th}$ is the thermal resistance.

Temperature changes causes shear thermal stress, while shear thermal stress causes liftoffs and cracks [6]. Due to the encapsulation, internal liftoffs and cracks cannot be observed directly; several sensitive parameters have been considered to represent the fatigue characteristics instead, such as the IGBT saturation voltage drop ($U_{CES}$) [7], SOA [8], thermal resistance between IGBT junction and device substrate ($R_{th}$ [9], threshold voltage of the IGBT gate [10], etc. These methods are based on the one-to-one relationship between fatigue signatures and fatigue levels. However, these parameters are difficult to obtain reliably in the situation of high power and high EMI (Electromagnetic Interference), such as converters.

In other cases, liftoffs and cracks can be estimated by experimental data where a certain number of thermal cycles and Temperature drift (obtained from the mission profile) are applied to the IGBT under test; we call it AFT (accelerated fatigue test). Then, the model is built based on fatigue observations from the AFT. The model that is established by the AFT is the fitting relationship between the fixed working conditions and the fatigue life of switching devices.

An AFT tests a DUT with a predetermined time-effect force and obtains data showing the fatigue characteristics of it. However, the existing fatigue model requires stable and fixed operating conditions because these models were established under the AFT results of fixed experimental conditions [4,11,12]. Therefore, the existing model cannot be fully applied to the changing conditions of the actual operating conditions, and this is the actual application of the TC. For example, in practical TC operation, the switching device is affected by the changing ambient temperature due to the movement of the EMU, while the ambient temperature remains constant in AFT [13]; in addition, the switching frequency of the switching device (i.e., The number of currents turn-ons and turn-offs in one second) and currents are also constantly changing as the speed of the EMU or the passenger load changes, however the conventional AFT only considers the constant switching frequency and the constant test current [14].

In this paper, it shows that Weibull distribution is more suitable to describe the cumulative failure rate of high-power switching device fatigue based on the data from conventional AFT. Then, a novel bidirectional aging approach is proposed to show the aging effect from FWD inside the switching device. In the process of AFT, the experiment time is often too long under low current. Therefore, a weighted LSM (least squares method) is proposed to complete the piecewise fitting of BAFT (bidirectional AFT) experimental data, which improves the accuracy of modeling. Through the aging model that is proposed in this paper, the service life of switching devices under actual working conditions can be predicted and the fault trend can be evaluated online when the load current of switching devices is known.

## 2. Fatigue Characteristics of High-Power IGBT Model

### 2.1. Analysis of the Results of Conventional Single-Directional Aging Experiments

The analytic model of the IGBT is the analytical relationship between the sensitive factor in the IGBT and the number of failures (i.e., the lifetime). The aging data is fitted correspondingly, and the expression is intuitive. In order to describe the relationship between the cumulative failure rate *F* and the number of experiments (i.e., the number of cycles), it is necessary to select an appropriate life distribution. If the distribution model is properly selected, the *F* under various working conditions could be obtained with more accuracy [15]. Common life distribution models include normal distribution, lognormal distribution, Weibull distribution and log-Weibull distribution. The choosing of distribution model is the key to switching device fault reconfiguration.

The normal distribution, also known as the Gauss distribution, is the most common and widely used distribution of the probability distribution of all random phenomena. It can be used to describe many natural phenomena and various physical properties. In practical applications, there are many experimental data that can be fitted with a normal distribution as its approximate distribution [16]. The lognormal distribution is a typical model that is derived from a normal distribution. When the logarithm of the expiration time "*ln* (*t*)" obeys the normal distribution, *t* obeys logarithmic normal distribution. Logarithmic transformation can reduce the larger number to a smaller number. Lognormal distribution is widely used in fatigue life research [17]. The Weibull distribution is a continuous distribution that is widely used in reliability, and its advantage lies in its strong adaptability to various types of test data. It is widely used in reliability engineering, especially for the distribution of wear and tear failure of electromechanical products. Since it can easily infer its distribution parameters using probability values, it is widely used in data processing of various life tests [15]. The log-Weibull distribution is a special correlation distribution of the Weibull distribution which is obtained by taking the logarithm of *x* and obeying the Weibull distribution. The effect is similar to the lognormal distribution, and the larger number can be reduced by logarithmic transformation [18].

Table 1 is the equation for the distribution function and density function of the four distributions.

**Table 1.** Distribution functions and density functions of the four distributions.

| Name | Distribution Function | Density Function |
|---|---|---|
| Normal distribution | $F(x) = \frac{1}{\sigma\sqrt{2\pi}} \int_{-\infty}^{x} e^{-\frac{(t-\mu)^2}{2\sigma^2}} dt$ | $f(x,\mu,\sigma) = \begin{cases} \frac{1}{\sigma\sqrt{2\pi}} e^{-\frac{(x-\mu)^2}{2\sigma^2}} & ,x>0 \\ 0 & ,x\leq 0 \end{cases}$ |
| Lognormal distribution | $F(x) = \frac{1}{\sigma\sqrt{2\pi}} \int_{-\infty}^{\ln x} e^{-\frac{(t-\mu)^2}{2\sigma^2}} dt$ | $f(x,\mu,\sigma) = \begin{cases} \frac{1}{\sigma x\sqrt{2\pi}} e^{-\frac{(\ln x-\mu)^2}{2\sigma^2}} & ,x>0 \\ 0 & ,x\leq 0 \end{cases}$ |
| Weibull distribution | $F(x) = \begin{cases} 1 - e^{-\left(\frac{x}{\lambda}\right)^k} & ,x>0 \\ 0 & ,x\leq 0 \end{cases}$ | $f(x,k,\lambda) = \begin{cases} \frac{k}{\lambda}\left(\frac{x}{\lambda}\right)^{k-1} e^{-\left(\frac{x}{\lambda}\right)^k} & ,x>0 \\ 0 & ,x\leq 0 \end{cases}$ |
| Logarithmic Weibull distribution | $F(x) = \begin{cases} 1 - e^{-\left(\frac{\ln x}{\lambda}\right)^k} & ,x>0 \\ 0 & ,x\leq 0 \end{cases}$ | $f(x,k,\lambda) = \begin{cases} \frac{k}{\lambda x}\left(\frac{x}{\lambda}\right)^{k-1} e^{-\left(\frac{\ln x}{\lambda}\right)^k} & ,x>0 \\ 0 & ,x\leq 0 \end{cases}$ |

Because the design life of IGBTs is generally more than 20 years, aging data is generally obtained by means of accelerated fatigue test (AFT) [19].

In Figure 2, $G_1$ is auxiliary GTO, $I_1$ is a fatigue current and $I_2$ is used to calculate $T_j$. R and C form an absorption branch. If $G_1$ is turned on, $I_1$ flows through the IGBT part. If $G_1$ is turned off, $G_2$ provides the freewheeling route of $I_1$. $C_1$ and $C_2$ produce $I_1$ and $I_2$, respectively. During the experiment, aging current $I_1$ only flows through IGBT. So, it can be called SAFT.

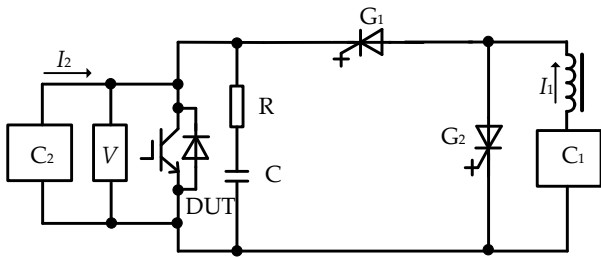

**Figure 2.** Schematic diagram of single-directional accelerated fatigue test.

During AFT, the ambient temperature is approximately 25 °C. $I_2$ is 1.5 A, which is no more than 0.1% of the DUT rated current (1500 A) as a rule of thumb. The cycle time is 12 s with a duty cycle of 48%. During the AFT, DUT is placed in the actual TC power module and is then connected to the test platform. In AFT, $F$ is calculated from the observed $U_{CES}$ [7], as shown in Equation (3).

$$F = \frac{U_{CESat} - U_{CESat0}}{U_{CESat0}} \tag{3}$$

In Equation (3), $U_{CES0}$ is the initial saturation voltage of a new DUT, and $U_{CES}$ is the current value. When $U_{CES}$ rises to 1.2 times that of $U_{CES0}$, $F$ is defined to be 1.

With the data from SAFT, the corresponding $F$ is calculated. Four different distribution models are fitted by the LSM method. As for the normal distribution, it gives:

$$F(x) = \frac{1}{35500\sqrt{2\pi}} \int_{-\infty}^{x} e^{-\frac{(t-309254)^2}{2\times 35500^2}} \, dt, \mu = 309254, \sigma = 35500 \tag{4}$$

As for the lognormal distribution, it gives:

$$F(x) = \frac{1}{0.127\sqrt{2\pi}} \int_{-\infty}^{\ln x} e^{-\frac{(t-12.665)^2}{2\times 0.127^2}} \, dt, \mu = 12.665, \sigma = 0.127 \tag{5}$$

As for the Weibull distribution, it gives:

$$F(x) = 1 - e^{-(\frac{x}{317254})^{12.98}}, \lambda = 317254, \ k = 12.98 \tag{6}$$

As for the Logarithmic Weibull distribution, it gives:

$$F(x) = 1 - e^{-(\frac{\ln x}{165})^{12.667}}, \lambda = 165, \ k = 12.667 \tag{7}$$

In order to select the most suitable one of the four distributions, we plot the curve of the experimental data and four distributions, and the density functions are also plotted. The results are shown in Figures 3 and 4. It should be noted that in Figures 3 and 4, the curve of the Weibull distribution coincides with the curve of the Logarithmic Weibull distribution. In Figures 3 and 4, the horizontal axis is the number of cycle times, and the vertical axis is cumulative failure rate, which is calculated according to DUT saturation voltage drop [13,17].

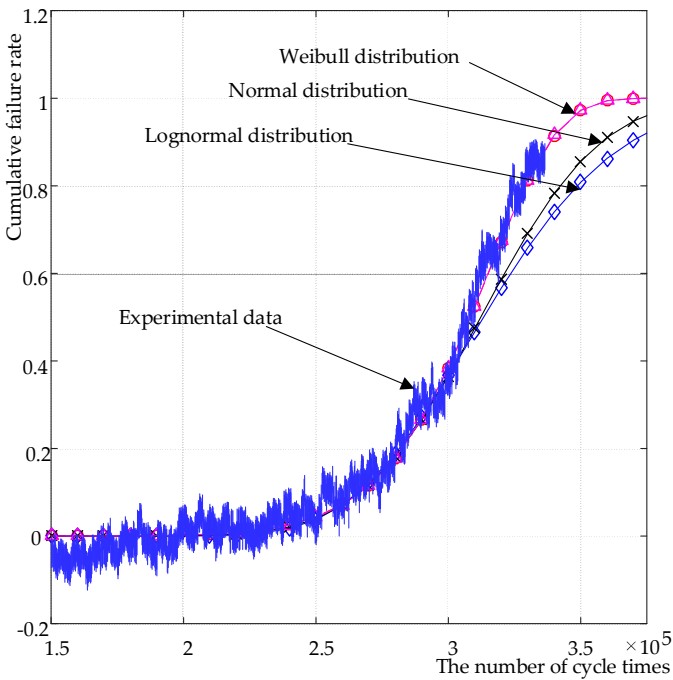

**Figure 3.** Comparison between data curves of single-direction 1500 A fatigue test and four distribution fitting results.

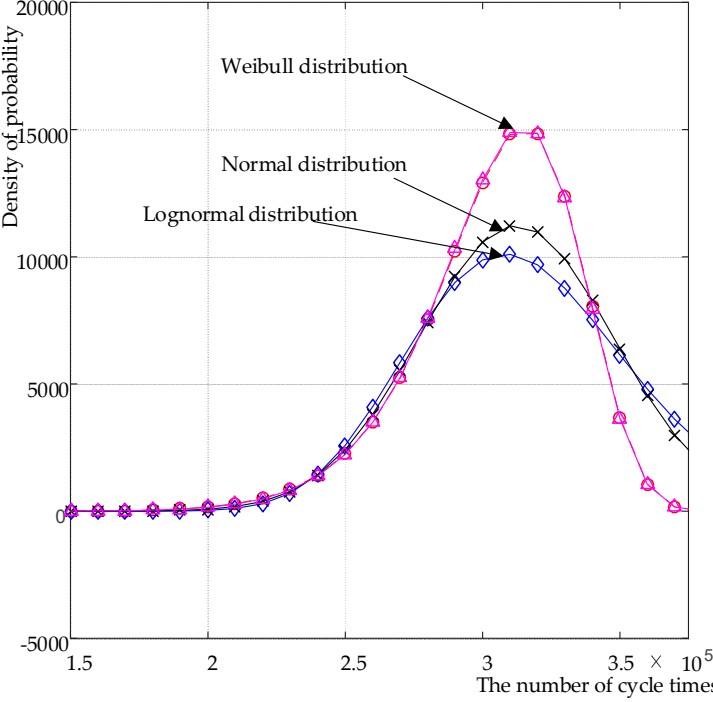

**Figure 4.** Comparison of the density function of Weibull distribution, Logarithmic Weibull distribution and normal distribution, lognormal distribution under the single-direction 1500 A fatigue test.

From Figure 4, it seems that the Weibull distribution is closer to the practical fatigue process (hence the fault transition process) compared with the normal distribution, because the latter one shows a larger error. What is more, the lognormal distribution is basically consistent with normal distribution; hence the same fitting accuracy is obtained. However, the transient component of failure density in the case of the lognormal distribution function is too high; therefore, it is not suitable to describe the failure density of high-power IGBTs. The logarithmic Weibull distribution has the same

characterization accuracy as the Weibull distribution, however its calculation process is much more complicated. Finally, the Weibull distribution is the most suitable one.

## 2.2. Effect of FWD Actions on Device Aging

It is not enough to consider IGBT only in the fatigue model. In the device, current can flow in two directions, as shown in Figure 5, where C and E are the external terminals of the device. In Figure 5a, current flows through IGBT. In Figure 5b, current flows through FWD. In AFT, the DUT is heated by the test current flowing through it. However, conventional AFT only offers the test current in one direction, i.e., the direction shown in Figure 5a [13–15], considering that IGBT is more fragile. In conventional AFT (hence SAFT, single-directional AFT), the aging of IGBT is equal to the aging of DUT. However, the current through FWD also generates heat, hence FWD actions also cause DUT to age. SAFT should be improved to bidirectional AFT (BAFT) where the test current flows through both the IGBT and FWD. The data that is obtained by BAFT accurately shows the aging effects caused by IGBTs and FWD. Such an effect is shown more clearly in junction temperature variation, as shown in the simulation of the TC operation below.

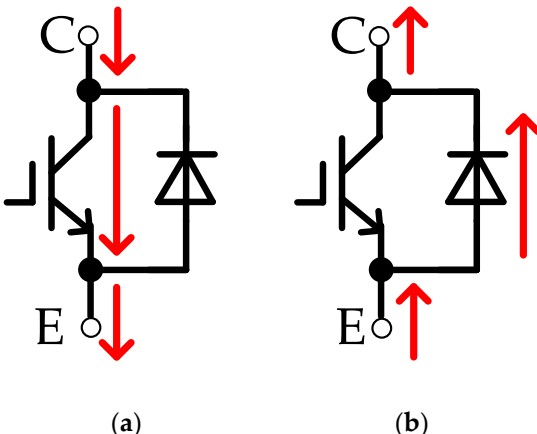

(**a**)  (**b**)

**Figure 5.** IGBT and FWD current flow differences. (**a**) IGBT current flow; (**b**) FWD current flow.

Figure 6 shows TC topology, where $Q_{11} \sim Q_{32}$ are IGBTs, $D_{11} \sim D_{32}$ are FWDs, QB, RB and DB constitute the brake energy consumption branch. $Q_{11} \sim Q_{32}$ and $D_{11} \sim D_{32}$ are usually packaged in six modules onto the heat sink. QB and DB is not used frequency, therefore only the fatigue of $Q_{11} \sim Q_{32}$ and $D_{11} \sim D_{32}$ are considered here.

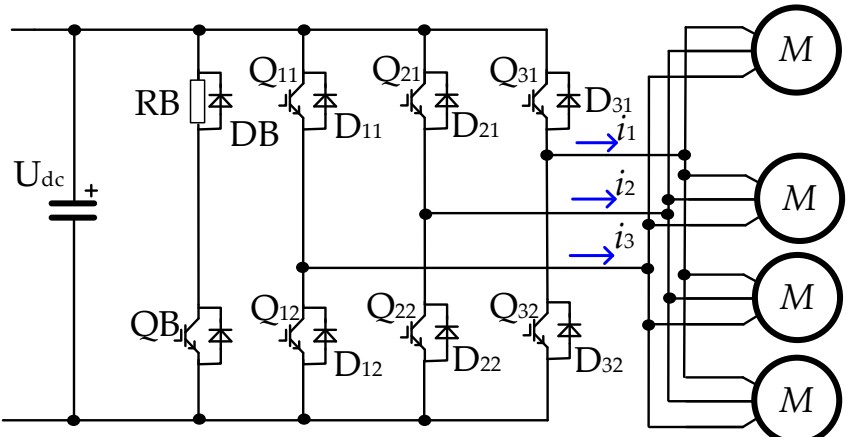

**Figure 6.** TC (traction converter) circuit topology.

The internal temperature of the IGBTs and FWDs under the given operating conditions was obtained in an electro-thermal simulation of the TC model. In the simulation, the EMU went through two operating cycles. The simulation results are shown in Figure 7. The horizontal axis in Figure 7 is time, and the vertical axis is junction temperature. The switching device used is 1500 A/3300V, the switching frequency is 57 Hz~1000 Hz and the ambient temperature is 25 °C.

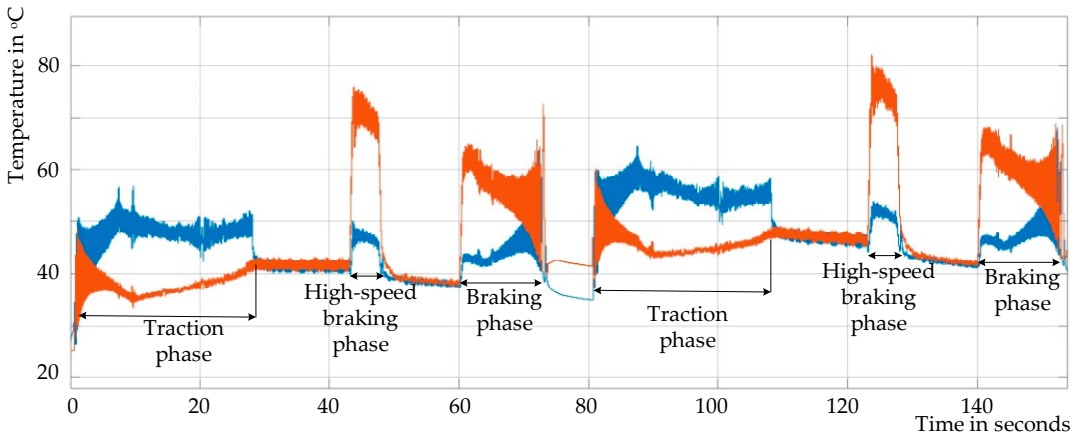

**Figure 7.** Junction Temperature Simulation of $Q_{32}$ and $D_{32}$ under AW0.

In Figure 7, the maximum variation of FWD junction temperature ($\Delta T_{j2}$) is 56.3 °C, and that of IGBT ($\Delta T_{j1}$) is 39.7 °C. Junction temperature variation severely affects device lifetime [20,21]. According to the experimental data of LESIT [22] and CIPS08 [23], the junction temperature change and the average junction temperature are the key factors affecting the aging process. Table 2 shows the maximum values of $\Delta T_{j2}$ and $\Delta T_{j1}$ under different load conditions [24]. $\Delta T_{j2}$ and $\Delta T_{j1}$ are affected by various factors, such as power loss of IGBT and FWD, heat dissipation condition and thermal resistance. In Table 2, $\Delta T_{j2}$ and $\Delta T_{j1}$ increase as the vehicle load increases (which appears in the form of load current, i.e., the fatigue current), which means that there is certain one-to-one correspondence between $\Delta T_{j1}$, $\Delta T_{j2}$ and the load current.

**Table 2.** $\Delta T_j$ of $Q_{32}$ and $D_{32}$ under different load conditions.

| Load Condition | $\Delta T_{j1}$ | $\Delta T_{j2}$ |
| --- | --- | --- |
| AW0 | 39.7 °C | 56.3 °C |
| AW2 | 52.6 °C | 72.1 °C |
| AW3 | 60.9 °C | 87.2 °C |

In Figure 7, it should be noted that $\Delta T_{j1}$ is higher in the traction phase than $\Delta T_{j2}$ and lower in the braking phase. What is more, $\Delta T_{j2}$ grows much faster than $\Delta T_{j1}$ in the high-speed braking phase. This stems from the fact that more TC power output is required in the high-speed braking phase than in the traction phase, and such braking power mainly flows through the FWD instead of the IGBT [6]. Therefore, the role of FWD should not be ignored during the fatigue of the switching device.

As stated above, the IGBT ages faster with the FWD actions, i.e., the fatigue process is accelerated. If the aging state of the DUT is still represented by the aging state of the IGBT (as is the case in the existing references [2–4]), this acceleration effect of the FWD action must be considered.

### 2.3. BAFT and the Analysis of BAFT Results

The fatigue model reveals the relationship between the main factors affecting fatigue accumulation and the degree of fatigue. The load condition of the vehicle directly affects the fatigue of the switching device. The TC current output reflects the load level and this current flow completely through the switching device. Therefore, there is a one-to-one correspondence between the TC current output

and the fatigue level of the switching device, and there is a one-to-one correspondence between the device current and the fatigue level and between the device current and the device lifetime. The correspondence is shown in Equation (8).

$$N_f = f(I_{eq}) \tag{8}$$

$N_f$ is the number of cycles the device can withstand before failure. We denote $N_f$ as the lifetime of the device; $I_{eq}$ is the equivalent current amplitude of the device.

The acceleration effect of the FWD action means that the life of the device is shortened, as shown in Equation (9).

$$N_{f2} = nN_{f1} \tag{9}$$

In Equation (7), $N_{f2}$ is the practical lifetime with FWD action, while $N_{f1}$ is the lifetime of only considering IGBT action; $n$ is a coefficient indicating an acceleration effect.

However, $n$ is difficult to obtain through experimental testing, so other ways need to be considered to express the aging acceleration effect of FWD on power modules. For better applicability, we present the Equation (8) in its current form, as shown in Equation (10).

$$I_{eq}^2 = i_G^2 + \mu^2 i_D^2 \tag{10}$$

In Equation (10), $i_G$ is the current of the IGBT, $i_D$ is the current of FWD and $\mu$ is the acceleration factor. The fatigue of semiconductor devices is caused by the accumulation of thermal stress effects, so the current is in the form of RMS (Root-Mean-Square), and there is a square operation.

In order to obtain $\mu$, we propose a new AFT (BAFT) with bi-directional fatigue current. The BAFT test platform is shown in Figure 8.

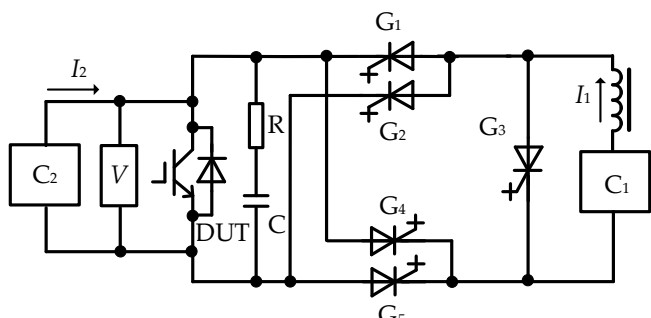

**Figure 8.** Schematic diagram of a bi-directional accelerated fatigue test.

In Figure 8, the device functions are similar to those in the single-directional accelerated fatigue test platform of Figure 2, except that when $G_2$ and $G_4$ are turned on, $I_1$ flows through the FWD portion inside the DUT. In BAFT, the number of cycles of IGBT and FWD is equal, so only half of the total number of actions is considered.

For comparison, a bi-directional experiment was performed at $I_1$ of 1500 A. Based on the data obtained from the bi-directional experiment, the cumulative failure rate curve of BAFT is shown together with the cumulative failure rate curve of SAFT at 1500 A in Figure 9. In Figure 9, the horizontal axis is the number of cycle times and the vertical axis is cumulative failure rate. It can be seen that the bi-directional experiment significantly accelerates the aging of the entire switching device.

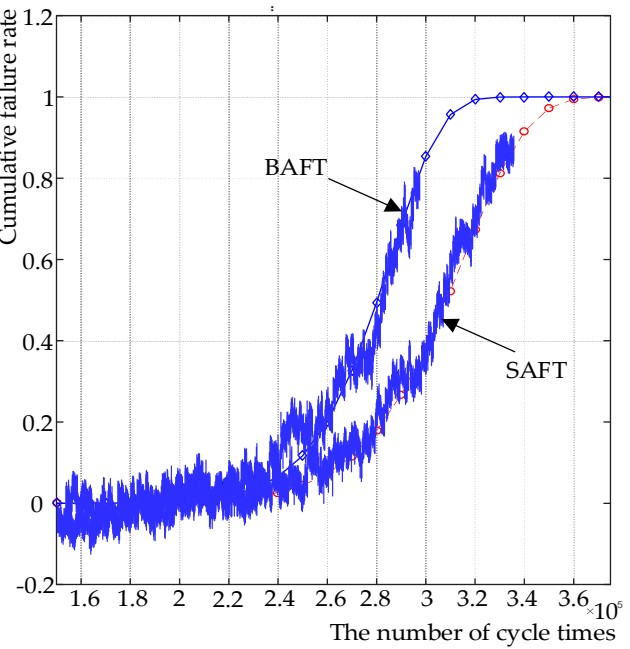

**Figure 9.** BAFT (bidirectional AFT) and SAFT curves obtained under 1500 A.

Since the fatigue process of the switching device follows the Weibull distribution, the Weibull distribution is fitted by the least squares method according to the bi-directional aging experimental data under 1500 A, and the mathematical analysis relationship between the BAFT and the number of cycles $x$ can be fitted to:

$$F_{\mathrm{B}}(x) = 1 - e^{-\left(\frac{x}{283769}\right)^{15.05}} \tag{11}$$

The result curve of SAFT is fitted as:

$$F_{\mathrm{S}}(x) = 1 - e^{-\left(\frac{x}{317254}\right)^{12.98}} \tag{12}$$

The mean of the Weibull distribution:

$$E = \lambda\Gamma(\frac{1}{k} + 1) \tag{13}$$

In Equation (13), $\Gamma$ is a gamma function, and $\lambda$ and $k$ are parameters of the Weibull distribution.

To derive $\mu$, we define an intermediate variable $m$ which is derived from the quotient between the mean of $F_{\mathrm{B}}(x)$ and $F_{\mathrm{S}}(x)$, as shown in Equation (14).

$$m = \frac{\lambda_2\Gamma(1 + \frac{1}{k_2})}{\lambda_1\Gamma(1 + \frac{1}{k_1})} \tag{14}$$

In Equation (14), $\lambda_1$ = 283769, $k_1$ = 15.05, $\lambda_2$ = 317254, $k_2$ = 12.98. Using these values, $m$ can be calculated as 1.1125.

From the difference between BAFT and SAFT, it is given that:

$$\begin{cases} I_{\mathrm{eq}} = qi_{\mathrm{G}} \\ i_{\mathrm{G}} = i_{\mathrm{D}} \end{cases} \tag{15}$$

Consider $i_{\mathrm{G}} = i_{\mathrm{D}}$ = 1500A and $m$ = 1.1125. Equation (10) shows that FWD accelerates the aging speed by 23.77%.

*2.4. Fault Model Fitted by Weighted LSM*

However, it is not enough to analyze the experimental data of bi-directional aging under a single current. In order to obtain a more consistent distribution function with the practical trend of failure rate, this paper will analyze the bi-directional aging under different current values. Take the bi-directional 900A experiment as an example. The data collected from the experiment are plotted in Figure 10. In Figure 10, the horizontal axis is the number of cycle times and the vertical axis is cumulative failure rate.

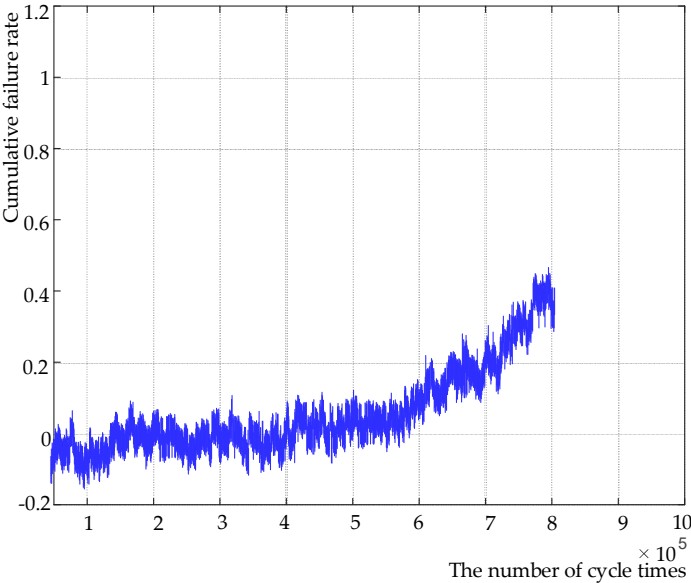

**Figure 10.** Curve of bi-directional aging under 900A.

It should be noted that when the test current decreases, the aging speed decreases sharply [25]. It is too difficult to obtain complete aging data through experiments. Therefore, only the part with a failure rate of 0.4 is used for analysis.

Through the previous analysis, Weibull distribution is selected to fit the failure rate distribution. The least squares method can be used to estimate the parameters of non-linear regression analysis. The traditional methods of least squares are direct search method, lattice search method, Gauss-Newton method, Newton-Raphson method, etc. [26]. Direct search method and lattice search method are inefficient and seldom used in practical engineering [27]. Gauss-Newton method is also called linearization method. Newton-Raphson method is an improvement of Gauss-Newton method. However, in some cases, these two methods may require much iteration to converge, and in a few cases, they may not converge at all [26].

To a certain extent, the problem of the above method can be avoided by using a genetic algorithm for least squares. GA (genetic algorithm) is an adaptive stochastic search method that is based on natural selection and biogenetic theory [28]. Firstly, the solution of the problem to be solved is regarded as a number of the individuals in the population. The individual in the population is coded and expressed as a string. Then, according to the fitness of the individuals, the selection, crossing and variation operations between the individuals are simulated to generate new solutions. In the process of population cyclic evolution, individuals gradually evolve to the state of the approximate optimal solution; therefore, the optimal solution of the problem can be obtained. The flow chart is shown in Figure 11.

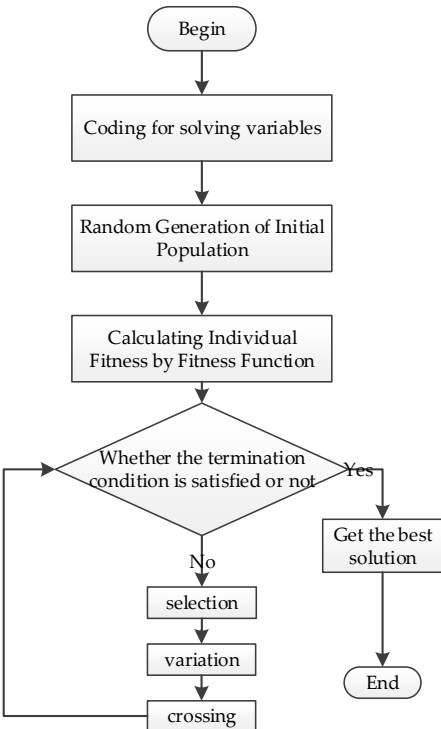

**Figure 11.** Flow chart of genetic algorithm.

The most important part of GA is fitness function. When GA simulates the selection of nature, the adaptation of each generation of population to conditions corresponds to a necessary parameter, which determines how much probability the individuals in this population participate in the heredity. The function of evaluating this probability is called fitness function. The suitability of fitness function selection will affect the whole GA. Selecting a different fitness function will result in different convergence and accuracy of the whole algorithm [29].

Aiming at the fitting of Weibull distribution in this paper, the purpose of our fitting is to find the appropriate values of $\lambda$ and $k$, so we take $\lambda$ and $k$ as variables in the initial population.

The fitness function selected is:

$$fitness = \sum_{i=1}^{n} \left(1 - e^{-\left(\frac{x_i}{\lambda}\right)^k} - Y_i\right)^2 \tag{16}$$

In Equation (16), $x_i$ is the number of experimental cycles and $Y_i$ is the failure rate corresponding to the number i experimental cycle in the actual experimental data.

It should be noted that GA is to solve the fitness function to a minimum value, and the corresponding variables of the minimum value as the output of the optimal solution. The concept of LSM is also used in the construction of fitness function. The purpose of the fitness function is to minimize the absolute value of the difference between the output value of the function to be fitted and the actual data. After summing the squares of the difference between each output value and the practical data, we can calculate a function that is closest to the curve that was formed by the practical data.

Because of the existence of crossing and variation operations, GA has randomness in the process of finding the optimal solution, which can avoid the algorithm falling into local optimum to a certain extent. The search for the optimal solution by GA is general. The fitting degree of $\lambda$ and $k$ fitted by GA should be the closest to all data points, and the fitted function should be the closest to the actual curve.

Taking $\lambda$ and $k$ as input variables and Equation (16) as fitness function, the algorithm is simulated by the GA toolbox of MATLAB. After 200 generations of the calculation, the optimal value is $\lambda = 925830$,

$k = 5.7629$. The relationship curve between the optimal solution of each generation and the number of iterations of the algorithm is shown in Figure 12. In Figure 12, the horizontal axis is the number of generation and the vertical axis is fitness value. It can be seen that the algorithm converges basically after about 10 generations.

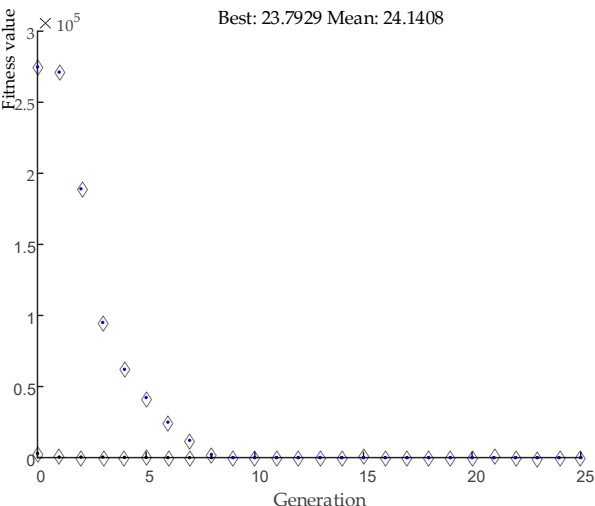

**Figure 12.** Optimal Solution Curve of Each Generation.

Using GA to calculate this can get better fitting accuracy and convergence, however when facing a large number of data in IGBT acceleration test, GA takes too much time, which is the biggest problem of genetic algorithm. Therefore, we introduce a weighted least squares method.

The weighted algorithm is helpful to select the most important region of a set of data. Because the aging process is too slow and the data that are obtained are less when the current is small, in order to describe the average level of bi-directional aging of IGBT under low current (900 A), the LSM method of weighted residual is introduced in this paper. The LSM method of weighted residuals can conveniently determine which interval in the whole definition domain is the most concerned.

The standard LSM is to form a hyper plane in two dimensions (single independent variable, single dependent variable or SISO) by some functions. The sum of the distances between the hyper plane and the points in the sample space is the smallest. This hyper plane can be defined as $span\{\varphi_1(x), \varphi_2(x)..., \varphi_n(x)\}$; the equation at any point on the extended hyper plane can be defined as:

$$f(x) = a_1 \bullet \varphi_1(x) + a_2 \bullet \varphi_2(x)... + a_n \bullet \varphi_n(x) \tag{17}$$

In fact,

$$y_i = f(x_i), (x_i, y_i) \in C \tag{18}$$

In Equation (17), C is the set of sample spaces.

In weighted LSM, we define residual as the sum of the distances between points and the hyper planes.

$$\Psi = \sum_{i=1}^{m} \|f(x_i) - y_i\|^2 \tag{19}$$

In the improved LSM, the definition

$$\Psi = \sum_{i=1}^{m} \alpha_i \|f(x_i) - y_i\|^2, \sum_{i=1}^{m} \alpha_i = 1 \tag{20}$$

where $\alpha_i$ is defined as the quotient between the number of abscissa $x_i \in (X_i, X_i + \Delta X)$ of sample point $(x_i, y_i)$ and the total number of sample points.

$\Delta X$ is usually determined by experiment or experience. For IGBT aging failure rate fitting, it is actually the resolution of junction temperature fluctuation (the relative relationship between failure rate and junction temperature fluctuation) or the time resolution (the relationship between failure rate and time).

$$\frac{\partial \Psi}{\partial a_h} = \frac{\partial}{\partial a_h} \sum_{i=1}^{m} \alpha_i \|f(x_i) - y_i\|^2$$

$$\frac{\partial \Psi}{\partial a_h} = \frac{\partial}{\partial a_h} \{ \sum_{i=1}^{m} \alpha_i [(\sum_{j=1}^{n} a_j \varphi_j(x_i))^2] \} - 2 \frac{\partial}{\partial a_h} \{ \sum_{i=1}^{m} [y_i \sum_{j=1}^{n} a_j \varphi_j(x_i)] \}$$

$$\therefore \frac{\partial \Psi}{\partial a_h} = 0 \Rightarrow \sum_{j=1}^{n} \{ a_j \sum_{i=1}^{m} [\alpha_i \varphi_h(x_i) \varphi_j(x_i)] \} = [\sum_{i=1}^{m} y_i \alpha_i \varphi_h(x_i)]$$

If defined $\sum_{j=1}^{n} [\alpha_i \varphi_h(x_i) \varphi_j(x_i)] = < \alpha\varphi_j, \varphi_h >$, $\sum_{i=1}^{m} y_i \alpha_i \varphi_h(x_i) = < \alpha f, \varphi_h >$

This is actually the definition of product in vector space.

$$\therefore \frac{\partial \Psi}{\partial a_h} = 0 \Rightarrow \sum_{j=1}^{n} [a_j < \alpha\varphi_j, \varphi_h >] = < \alpha f, \varphi_h >$$

$$\therefore \begin{cases} \frac{\partial \Psi}{\partial a_1} = 0 \\ \frac{\partial \Psi}{\partial a_2} = 0 \\ \dots \\ \frac{\partial \Psi}{\partial a_n} = 0 \end{cases} \Rightarrow \begin{pmatrix} < \alpha\varphi_1, \varphi_1 > & < \alpha\varphi_2, \varphi_1 > & \dots & < \alpha\varphi_n, \varphi_1 > \\ < \alpha\varphi_1, \varphi_2 > & < \alpha\varphi_2, \varphi_2 > & \dots & < \alpha\varphi_n, \varphi_2 > \\ \dots & \dots & \dots & \dots \\ < \alpha\varphi_1, \varphi_n > & < \alpha\varphi_2, \varphi_n > & \dots & < \alpha\varphi_n, \varphi_n > \end{pmatrix} \begin{pmatrix} a_1 \\ a_2 \\ \dots \\ a_n \end{pmatrix} = \begin{pmatrix} < \alpha f, \varphi_1 > \\ < \alpha f, \varphi_2 > \\ \dots \\ < \alpha f, \varphi_n > \end{pmatrix}$$

Note: Here A can be given according to the normal distribution.

According to the weighted LSM theory, we select different weights and obtain three curves, as shown in Figure 13. In Figure 13, the horizontal axis is the number of cycle times and the vertical axis is cumulative failure rate.

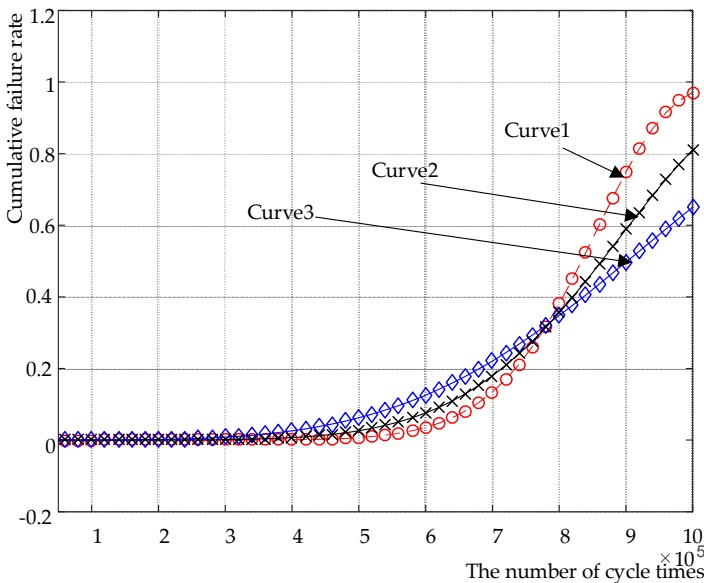

**Figure 13.** Weibull distribution function curves fitted under different weights.

Curve 1 is obtained by maximizing the weight of the interval $[7 \times 10^5, 8 \times 10^5]$ of experiments' number, parameter $\lambda = 868359$, $k = 8.99$, mean value calculated by Equation (5), $E_1 = 822222$.

Curve 2 is obtained by maximizing the weight of interval $[4 \times 10^5, 7 \times 10^5]$. The parameters $\lambda = 918264$, $k = 5.96$, $E2 = 851580$.

Curve 3 is obtained by maximizing the weight of interval $[1 \times 10^5, 5 \times 10^5]$. The parameters $\lambda = 987945$, $k = 4.02$, $E3 = 895730$. Compared with the experimental data curves, the three curves fit the original data most accurately in the interval with the largest weights.

From the point of view of the relationship between failure rate and time, in fact, what we need to focus on is not only the accuracy of characterization of failure rate in the whole life cycle of products, however also the accuracy of the most product failure processes. Accurately grasping the change of the latter is more conducive to accurately predicting the state of aging damage of components. According to this idea, observing the experimental data curve under 900 A, the failure rate rises rapidly in the interval where the number of tests is $[4.5 \times 10^5, 7.5 \times 10^5]$, so we think that curve 2 fits the aging situation under 900 A most accurately. By fitting the data with this weighted least squares method, the function corresponding to the above curve 2 is the closest to the actual distribution function.

In order to validate the distribution function fitted by weighted LSM, we compare the fitting results with those of GA. The function curve obtained is compared with the curve that was fitted by weighted LSM as shown in Figure 14, where the horizontal axis is the number of cycle times and the vertical axis is cumulative failure rate.

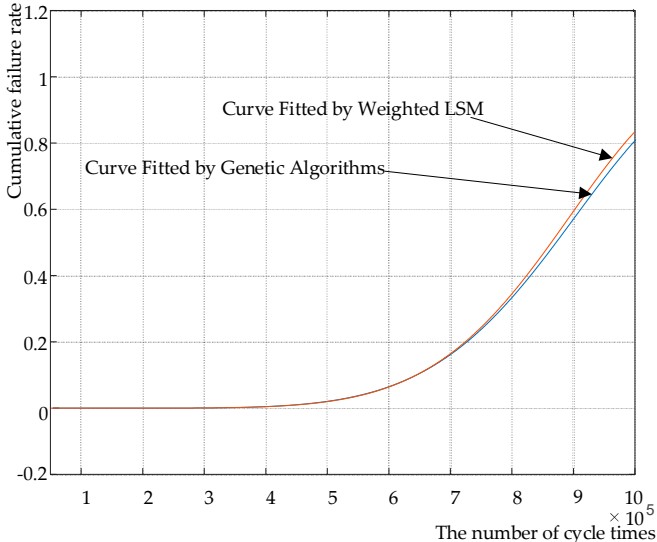

**Figure 14.** Comparison of Weibull distribution fitted by GA and curves fitted by weighted LSM (least squares method).

According to Figure 14, the curve of the function that was obtained by GA and curve 2 almost coincides in the interval of cycle number $[4 \times 10^5, 8 \times 10^5]$, which proves that the weighted LSM can more accurately fit the distribution function. Compared with the GA, the computational complexity of weighted LSM is greatly reduced and the computation time is shortened by 27% compared with the GA. So, the weighted LSM is more suitable for fitting the distribution.

### 2.5. Analytical Fatigue Model

Different BAFTs are based on different $I_1$, as shown in Figure 15. The horizontal axis in Figure 15 is the number of cycle times, and the vertical axis is cumulative failure rate. Under $I_1$ at 1200 A and 900 A, the mathematical model between cumulative failure rate $F$ and cycle number $x$ is as follows:

$$F_{B1200}(x) = 1 - e^{-\left(\frac{x}{508785}\right)^{8.01}} \tag{21}$$

$$F_{B900}(x) = 1 - e^{-\left(\frac{x}{918264}\right)^{5.96}} \tag{22}$$

Obtained from Equation (12)

$$F_{B1500}(x) = 1 - e^{-\left(\frac{x}{317254}\right)^{12.98}} \tag{23}$$

In Equations (21)–(23), $F_{B1500}(x)$, $F_{B1200}(x)$ and $F_{B900}(x)$ are the cumulative failure efficiencies of $I_1$ at 1500A, 1200A and 900A, respectively, and their mean values are 304890, 479170 and 851580. The corresponding equipment service life $N_f$ can be expressed in terms of the mean value.

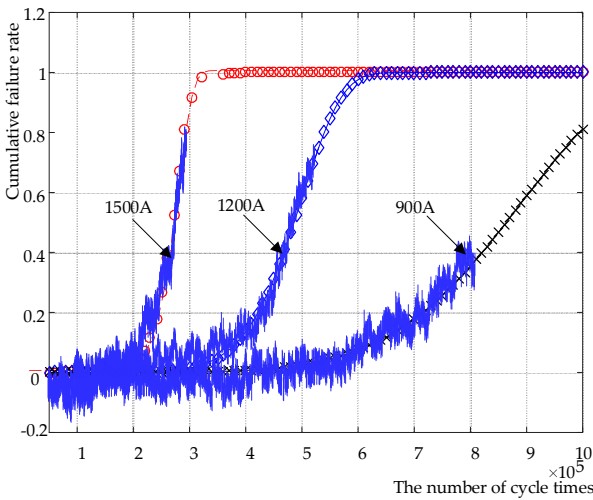

**Figure 15.** Curves obtained under different BAFT aging currents.

In addition, the ambient temperature $T_a$ should also be considered [30,31]. According to Equation (9), the $N_f$ expression considering $T_a$ is obtained.

$$N_f = f(I_{eq}) \times g(T_a) \tag{24}$$

In Equation (24), $g$ is specific functions.

Finally, after the fitting calculation, Equation (9) is expressed as:

$$N_f = (1.17I_1^2 - 3697.11I_1 + 3.23 \times 10^6) \times e^{k(\frac{1}{T_a} - \frac{1}{25})} \tag{25}$$

In Equation (25), $k$ is an Arrhenius coefficient. According to Arrhenius's law (a temperature increase of 10 °C means that the service life is shortened by half), $k$ is calculated to be 60.65. The experimental and fitting curves are shown in Figure 16.

Considering the RMS calculation scheme, the amplitude is $\sqrt{2}$ times of the effective value, so:

$$I_1 = \sqrt{2}I_{eq} \tag{26}$$

From Equation (26):

$$N_f = (2.34I_{eq}^2 - 5228.35I_{eq} + 3.23 \times 10^6) \times e^{60.65(\frac{1}{T_a} - \frac{1}{25})} \tag{27}$$

The data obtained in the experiment and the curve that was obtained by mathematical fitting is shown in Figure 16. The vertical axis in Figure 16 is the cycle counts that a DUT could survive before failure. The horizontal axis is the amplitude value of the test current. As shown in Figure 16, the obtained curve conforms to the obtained data very well.

By Equation (27), when $I_{eq}$ and $T_a$ are known, the service life or failure rate of the IGBT can be predicted. Equation (27) has been subjected to actual field data. The example we consider here is a metro line application in Shenzhen, a southern-China city, with the annual temperature average of 23 °C. The $I_{eq}$ has been recorded to be 489 A in a single year; therefore, $N_f$ is calculated to be 2,197,460. Considering that a train normally operates 204 cycles per day and normally 300 days per year, the train is capable of operating around 36 years under the same working conditions.

According to statistics offered by metro operating agency, the initial failure rate of a new module is 157 FIT. Given that they hold that the failure rate rise of 200% means that the switching devices in a batch have failed, the statistical service life of the applied power module in the specific metro line is around 47 years (offered by the agency). This means that the approach that was proposed has a prediction error of around 23%, which is within project tolerance (normally 20%~30%). It should be noted that the lifetime predicted by our model in Equation (27) is more likely to be true and more convincing, because a metro train is designed to operate for no more than 30 years, and the 47-year lifetime prediction does not take into consideration the acceleration of the aging process, as is shown by a bathtub curve.

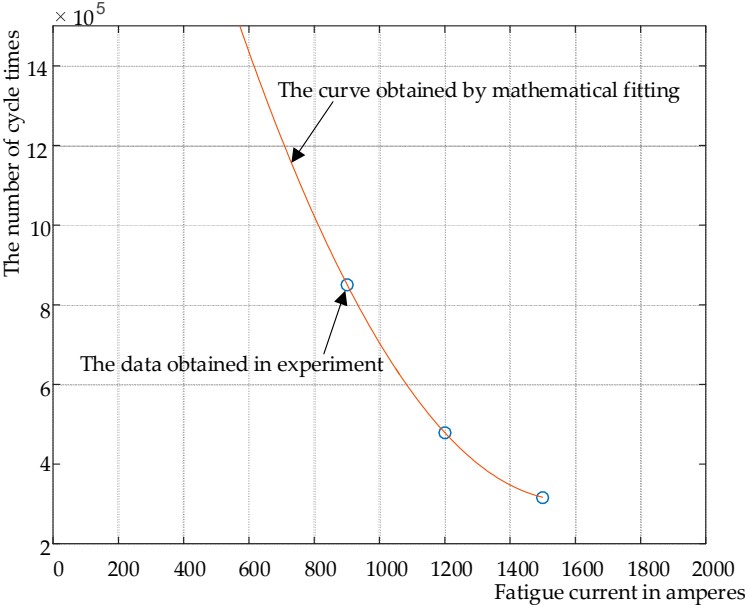

**Figure 16.** Experimental results of BAFT and fitting curves.

## 3. Conclusions

The object of the fault model in this paper is the fault model of the high-power switching device that was applied to rail transit. Based on a new test method using bidirectional current aging, we propose a method for extracting the acceleration factor of the freewheeling diode in the switching device for the entire module. Then, the weighting factor is introduced in the traditional least squares method, which improves the processing efficiency of the test results and the fitting accuracy of the model. Finally, an IGBT fault prediction model was presented by using the aging current and ambient temperature as sensitive factor.

**Author Contributions:** L.W. and S.L. designed the experiment, S.L. and L.W. performed the experiment, R.Q. did the data analysis, C.X. contributed analysis tools, L.W. wrote the paper.

**Funding:** This research was funded by the Fundamental Research Funds for the Central Universities, grant number 2018JBM060.

**Conflicts of Interest:** The authors declare no conflict of interest.

## Abbreviations and Nomenclature

| | |
|---|---|
| FWD | Free Wheeling Diode |
| EMU | Electric Multiple Unit |
| IGBT | Insulated Gate Bipolar Transistor |
| TC | Traction Converter |
| CTE | Coefficient of Thermal Expansion |
| DUT | Device Under Test |
| AFT | Accelerated Fatigue Test |
| GTO | Gate-Turn-Off thyristor |
| SAFT | Single-directional Accelerated Fatigue Test |
| BAFT | Bi-directional Accelerated Fatigue Test |
| EMI | Electromagnetic Interference |
| GA | Genetic Algorithm |
| S | Thermal stress |
| $R_{th}$ | Thermal resistance |
| $R_{th\text{-}c\_1}$ | Thermal resistance between the IGBT die and device case |
| $R_{th\text{-}c\_2}$ | Thermal resistance between the FWD die and device case |
| $\Delta CTE$ | Difference of CTE |
| $\Delta T$ | Junction temperature variation |
| $P$ | Heating power |
| $F$ | Cumulative failure rate |
| $I_1, I_2$ | the currents used in SAFT and BAFT |
| $U_{CES0}$ | Initial saturation voltage drop of the device |
| $U_{CES}$ | Saturation voltage drop of the device |
| $i_1, i_2, i_3$ | Current outputs of TC |
| $U_{dc}$ | DC voltage input of TC |
| $T_j$ | Junction temperature |
| $T_{j1}$ | Junction temperature of IGBT |
| $T_{j2}$ | Junction temperature of FWD |
| $\Delta T_{j1}$ | Junction temperature variation of IGBT |
| $\Delta T_{j2}$ | Junction temperature variation of FWD |
| $I_{eq}$ | Equivalent current amplitude through the device or fatigue current |
| $N_f$ | The service life of the device |
| $n, m$ | Coefficients to show the accelerating effect from FWD actions on device |
| $\mu$ | Acceleration factor |
| $i_G$ | IGBT current |
| $i_D$ | FWD current |
| $F_B(x), F_S(x), F_{B1500}(x),$ | the function relationship between F and the number of cycle times x under |
| $F_{B1200}(x), F_{B900}(x)$ | various conditions |
| $\lambda_1, k_1, \lambda_2, k_2$ | Parameters in Weibull distribution |
| $T_a$ | Ambient temperature |

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
