# Peer review of "Fault Prediction Model of High-Power Switching Device in Urban Railway Traction Converter with Bi-Directional Fatigue Data and Weighted LSM"

_applsci, doi:10.3390/app9030444_

Reviewer 1 Report

The paper is well written and the derivation of the fault prediction model is well presented. However, it is not clear if the derived model was verified or tested for its effectiveness and accuracy. The authors showed a comparison between the LMS and GA mthods in predeting the the districution fitting. Why the GA was selected for this comparison? Are there other methods that could also be compard with LMS? Give titles to tables!

Author Response

Point 1: The paper is well written and the derivation of the fault prediction model is well presented. However, it is not clear if the derived model was verified or tested for its effectiveness and accuracy.

Response 1:

The verification of proposed model has been added to section “2.5. Analytical fatigue model”.

The authors would like to express sincere thanks to the reviewer for his/her careful work.

The proposed model has been tested in Shenzhen Metro (Line 7), and all the statistical data given in the added part has been offered by the operating agency there. In the former version of the paper, the authors mainly try to introduce the model itself. However, taking the suggestion from the reviewer into consideration, the authors realize that it is much better and necessary to add the practical application example, in order to prove the effectiveness of the model proposed. Such part has been added just below Figure 16., in the last part of section 2.

In the example, it shows that the service life time which is predicted by the proposed model is more reasonable and realistic, and the predicted result is within project tolerance. As stated in the paper, “It should be noted that the lifetime predicted by our model in equation (27) is more likely to be true and more convincible, because metro train is designed to operate for no more than 30 years, and the 47-year lifetime prediction does not take into consideration the acceleration of aging process, as is shown by a bathtub curve.” The service life predicted by conventional approach is too optimistic and does not conform to the actual statistical data, while our predicted lifetime is more consistent with the actual situation on site.

Point 2: The authors showed a comparison between the LMS and GA mthods in predeting the the districution fitting. Why the GA was selected for this comparison? Are there other methods that could also be compard with LMS?

Response 2:

Existing approaches for nonlinear regression have been added into the revised version of the paper.

Existing approaches have been analyzed and referenced in the revised paper, such as direct search method, lattice search method, Gauss-Newton method, Newton-Raphson method, etc. Such part has been added below Figure10.

What’s more, GA has been applied into nonlinear regression in recent years with significant effectiveness and accuracy, but in our case of fatigue prediction (or fault prediction), the time consumption is too long for us to put it into field practice. That is why we compare our weighted LSM with GA. Such point has also be added into the revised version, below Figure 13.

Point 3: Give titles to tables! 

Response 3:

The title of Table 1 has been added

We are very sorry that the title of Table 1 has been omitted in the former version, and it has been added in the revised version. 

Reviewer 2 Report

No comments

Author Response

Point 1: No comments

Response 1:

The authors would like to express sincere thanks to the reviewer for his/her careful work. Several parts have been added into the revised version of the paper to offer verification of effectiveness of the proposed model, and to explain why weighted LSM is better than other conventional and existing algorithms in nonlinear regression in the case of switching device fault prediction. 

Reviewer 3 Report

All data graphs, except those in Figure 15 and 16, are unclear. Should add description of all axises.

Author Response

Point 1: All data graphs, except those in Figure 15 and 16, are unclear. Should add description of all axises.

Response 1:

The authors would like to express sincere thanks to the reviewer for his/her careful work. Explanation of the axis of Figure 9,10,11,13,14,15,16 has been added into the paper.

What’s more, several parts have been added into the revised version of the paper to offer verification of effectiveness of the proposed model, and to explain why weighted LSM is better than other conventional and existing algorithms in nonlinear regression in the case of switching device fault prediction.